# Reverse Remodeling and Functional Improvement of Left Ventricle in Patients with Chronic Heart Failure Treated with Sacubitril/Valsartan: Comparison between Non-Ischemic and Ischemic Etiology

**DOI:** 10.3390/jcm12020621

**Published:** 2023-01-12

**Authors:** Roberto Cemin, Simona Casablanca, Luisa Foco, Elisabeth Schoepf, Andrea Erlicher, Renato Di Gaetano, Davide Ermacora

**Affiliations:** 1Division of Cardiology, San Maurizio Regional Hospital, 39100 Bolzano, Italy; 2Eurac Research, Institute for Biomedicine, University of Lübeck, 39100 Bolzano, Italy

**Keywords:** chronic heart failure, ventricular function, sacubitril/valsartan

## Abstract

Sacubitril/valsartan (SV) has been demonstrated to reduce cardiovascular mortality, hospitalization for heart failure and to induce reverse ventricular remodeling. The present study was designed to confirm the effects of SV in a selected population of patients with HFrEF and to evaluate the different responses between patients with an ischemic or a non-ischemic etiology. A total of 79 patients with indication of SV were recruited prospectively during a timelapse of 4 years. SV was overall associated to a reduction of end-diastolic and end-systolic volume, of NT-proBNP levels, furosemide dosage and NYHA functional class, together with an increase in EF. These changes were more evident in patients with non-ischemic dilated cardiomyopathy, who showed a significant improvement in ventricular volumes, ejection fraction, TAPSE and blood levels of NT-proBNP. Kaplan–Meier curves confirmed a greater benefit in terms of ejection-fraction improvement in non-ischemic patients compared to the ischemic group. The results of the present study confirm the positive effect of SV on NYHA functional class, NT- proBNP, left ventricular volumes and EF in HFrEF patients, showing evidence of association of SV with ventricular remodeling in patients with dilated cardiomyopathy of non-ischemic etiology compared to the ischemic group.

## 1. Introduction

Heart failure is a clinical syndrome characterized by typical signs and symptoms caused by a structural and/or functional cardiac abnormality, resulting in reduced cardiac output and/or elevated intracardiac pressures at rest or during exertion [1]. In Western countries it affects about 1–2% of the adult population, with incidence and prevalence progressively increasing with age [2,3,4]. The recommended standard therapy in heart failure with reduced ejection fraction (HFrEF) should combine Ace-inhibitors (ACEi) or angiotensin receptor blockers (ARBs), beta blockers and aldosterone antagonists [1,4,5,6].

Recently the PARADIGM-HF study has showed an incremental reduction of CV mortality (−20%), all-cause mortality (−16%) and hospitalization for HF (−16%) in HFrEF patients treated with SV when compared to enalapril [7]. This effect was so evident that SV has been included in American and European guidelines with a class one recommendation for symptomatic HFrEF patients. In about a quarter of these patients, SV has been also associated with reverse remodeling [8], which is related to NT-proBNP reduction and better outcome [9,10,11,12,13,14]. 

CV mortality, all-cause mortality and hospitalizations for HF do not differ between the two etiological categories (ischemic and non-ischemic dilated cardiomyopathy), both in the group treated with enalapril and in SV [7,15]. Nonetheless, subsequent studies comparing non-ischemic and ischemic etiology have demonstrated a greater and significant improvement in ejection fraction in patients with non-ischemic cardiomyopathy, although long-term mortality remains comparable [16,17,18].

The aim of the present study was to confirm the positive effect of SV on NYHA functional class, NT-proBNP levels, furosemide dose, ventricular remodeling as well as on re-hospitalization events and cardiovascular death, also evaluating a possible different response in the population of patients with dilated cardiomyopathy of ischemic and non-ischemic etiology.

## 2. Materials and Methods

### 2.1. Patients

The patients were recruited prospectively and consecutively from the heart failure outpatient clinic of the San Maurizio Hospital of Bolzano between March 2017 and March 2020. They had a diagnosis of ischemic or non-ischemic dilated cardiomyopathy, were already on optimal medical therapy and had a clinical indication of SV. Inclusion criteria were: left ventricular ejection fraction ≤ 35%, NYHA functional class II or III and glomerular filtration rate (GFR) ≥ 15 mL/min, the best tolerated drug treatment with ACEi or ARB for at least 6 months, serum potassium levels < 5.4 mmol/L, systolic blood pressure values > 100 mmHg. Patients with concomitant right ventricular dilatation and/or dysfunction (4 patients), recent (<6 months) CRT implantation (6 patients) or concomitant dapagliflozin therapy (1 patient) were rolled out in a mean timelapse of 5 months from the beginning of the study (Figure 1). All the paced patients were on biventricular stimulation (CRT).

Before starting SV, all the patients underwent a complete transthoracic echocardiographic study, performed by the same trained operator with Philips Epiq 7q echocardiograph (Philips Healthcare, Andover, MA, USA). Baseline measurements included left ventricular end diastolic (EDV) and end systolic volume (ESV), ejection fraction (EF), left atrial volume (LA), an adequate evaluation of diastolic function (E and A velocities, E/A ratio, e’, E/e’ ratio), tricuspid ring systolic excursion (TAPSE). Diastolic dysfunction was evaluated according to the consensus document of Nagueh et al. [19]: diastolic dysfunction grade I (E/A ≤ 0.8 + E ≤ 50 cm/s), grade II (2 of 3 or 3 of 3, of the following criteria: average E/e’ > 14, TR velocity > 2.8 m/s, LA vol. > 34 mL/m^2^), grade III (E/A > 2). Degree of mitral insufficiency were classified as mild, moderate, or severe and volumes were indexed for body surface area. 

All the patients were commenced on 49/51 mg bid SV, with a dose reduction to 24/26 mg bid in case of significant glomerular filtration impairment (GFR ≤ 30 mL/min/1.73 m^2^), low systolic blood pressure (SBP ≤ 100 mmHg) or previous treatment with low doses of ACEi or ARBs. SV was started 36 h after last ACEi or 24 h after last ARBs intake. SV dose was doubled every 2–4 weeks up to an optimal target of 97/103 mg bid, if tolerated. In case of hypotension (symptoms or SBP ≤ 95 mmHg), hyperkaliemia or renal function worsening, an adjustment of the concomitant therapy was firstly attempted, followed by a temporary reduction of SV. Only in limited cases SV was definitively stopped. Clinical status, therapy, heart and renal function, electrolytes and NT-proBNP were regularly checked. Basal echocardiographic parameters (Table 1) were compared to those obtained during the last follow up visit. The patients gave their written informed consent to the study, which was conducted following the rules of the Declaration of Helsinki [20].

### 2.2. Study Endpoints

Primary endpoints were:Favorable (reverse) LV remodeling, classified as a reduction in EDV by more than 10%.Functional LV improvement (classified as a more than 10% increase of EF).A combined endpoint of “favorable LV remodeling” associated with “functional LV improvement” intended as a reduction of EDV by more than 10% associated to simultaneous improvement in EF of more than 10%.TAPSE improvement (>10% of basal value).Reduction in NT-proBNP (>10% of basal value).Improvement of the NYHA functional class.

Secondary endpoints were “cardiovascular death or hospitalization for heart failure” and furosemide dosage reduction by at least 10%. 

### 2.3. Statistical Analysis

We inspected data using classic descriptive statistics (frequencies, means, standard deviations, medians, interquartile ranges). As part of the initial exploratory analysis, clinical information between the ischemic and non-ischemic groups using Fisher’s exact test for categorical variables and the *t*-test or Wilcoxon test for continuous variables were compared. To analyze the different variables within subjects between baseline and last follow-up, the same tests in their version for paired data were used. The dependency of the differences between follow-up vs baseline variables in ischemic and non-ischemic patients was tested fitting a linear regression model with robust error estimation, correcting for age and gender. A time-to-event analysis, plotting Kaplan–Meier survival curves and log rank test was also performed and a Cox regression model was built, correcting for age and gender. A nominal significance level alpha = 0.05 was considered, without adjusting for multiple comparisons. For this reason, our analyses currently have an exploratory purpose.

Intra-observer variability, assumed as the average of all the relative intra-observer variability, was also checked for EF and EDV and calculated according to the formulas [21]: $\frac{\left|\mathrm{EF}1-\mathrm{EF}2\right|}{\left(\mathrm{EF}1+\mathrm{EF}2\right)/2}\times 100$; $\frac{\left|\mathrm{EDV}1-\mathrm{EDV}2\right|}{\left(\mathrm{EDV}1+\mathrm{EDV}2\right)/2}\times 100$ on a randomized sample of 30 patients. Agreement between the measures was assessed using Lin’s correlation coefficient and graphically represented using Bland–Altman plots.

## 3. Results

A total of 79 patients with a mean age of 68 (SD = 12) years were enrolled; 81% were male, 42/79 (53%) had an ischemic and 37 (47%) a non-ischemic etiology. From 90 patients evaluated at the beginning of the study, 11 were rolled out because of exclusion criteria. Eleven patients discontinued sacubitril/valsartan (SV) before the end of the study because of heart transplantation (one patient), death (six patients), ARI (one patient) and symptomatic hypotension (three patients). We also included these 11 patients in the statistical analysis, considering the event that led to drug suspension as the last follow-up date.

As regards the death event, this involved five patients belonging to the ischemic group and one patient belonging to the non-ischemic group with an average time between 225 and 769 days from the start of sacubitril valsartan.

Moreover, 15 hospitalizations for HF happened in the study timelapse (53% in the non-ischemic group). Enrollment and characteristics of population are shown in Figure 1 and Table 1. 

The ischemic group showed a baseline difference in some variables when compared to the non-ischemic one: male sex (93% vs. 68%; *p* = 0.01), NYHA III class (79% vs. 54%; *p* = 0.03) and grade III diastolic dysfunction (48% vs. 19%; *p* = 0.019) were more represented. Groups were homogeneous in terms of optimal medical therapy and percentage of ICD or CRT carriers, in line with the current literature. There were no patients on ivabradine therapy.

Creatinine values were greater in the ischemic group (1.19 vs. 1.04 mg/dl; *p* = 0.01), whilst baseline echocardiographic data were similar (Table 1). Intra-observer variability was 3.7% (SD = 3%) for EF and 6.3% (SD = 3.3%) for EDV.

Mean follow-up was 413 days with 43% of patients reaching the 97/103 mg bid dose and 36% the 49/51 mg bid. Twenty percent of the patients maintained the starting dose of 24/26 mg bid. None of the 79 patients experienced angioedema or hospitalization for acute myocardial infarction. No differences between the ischemic and the non-ischemic group were observed for NYHA class, new onset of atrial fibrillation (AF), hospitalizations for HF, death from all causes, cardiovascular death, highest SV dose reached and also for SV discontinuation because of acute renal impairment (ARI), hypotension, death or transplantation. Occasional withdrawal because of hypotension and hyperkaliemia was also similar. A more marked worsening of renal function (classified as a drop in GFR below 30 mL/min/1.73 m^2^ or a 50% reduction from baseline GFR) was observed in the group of ischemic patients (28% vs. 8%; *p* = 0.02).

A NYHA functional class improvement was seen in 56% of patients (*p* = 0.0002) with no significant differences between the two groups. The degree of diastolic dysfunction remained stable in 75% of non-ischemic patients and in 64% of ischemic patients, improved in 22% of non-ischemic and 26% of ischemic patients and worsened only in 3% of non-ischemic patients and in 10% of ischemic patients (Appendix A).

### 3.1. Differences of Relevant Clinical Parameters from Baseline

To investigate the effect of SV, the differences (“delta”) of the most important clinical and echocardiographic parameters between baseline and follow-up were calculated, and a comparison within patients and groups using the Wilcoxon matched test was performed (Table 2). 

In the entire sample, a reduction in EDV (−15 mL, *p* = 0.0002), ESV (−16 mL, *p* = 5.64 × 10^−6^), NT-proBNP (−538 pg/mL; *p* = 0.0159), furosemide dosage (−10 mg; *p* = 0.0221) and an increase in EF of 3% (*p* = 0.0002) were observed. 

There was a significant decrease in SBP (−7 mmHg; *p* = 9.23 × 10^−6^) and GFR (−3 mL/min/1.73 m^2^; *p* = 0.0439), but no evidence of significant changes of the E/e’ ratio and TAPSE.

The non-ischemic patients showed a significant improvement not only in EDV (−20 mL, *p* = 0.0004), ESV (−23 mL; *p* = 4.00 × 10^−6^) and EF (+6%; *p* = 3.26 × 10^−5^), but also in TAPSE (+1.2 mm, *p* = 0.0009), with a confirmed reduction of NT-proBNP (−571 pg/mL; *p* = 0.02).

The ischemic group did not show any significant difference between baseline and follow-up in echocardiographic and biochemical data but experienced a reduction in furosemide dose (−15 mg, *p* = 0.018, Appendix A). This reduction was not evident in the group of non-ischemic patients, who were however on a significantly lower baseline dosage. 

In order to test the association between the type of the disease and differences in endpoints at follow-up, as a control for age and sex a linear regression was performed (Table 3).

A significant improvement in EF of 6% (95% CI 2–9; *p* = 0.001) and in TAPSE of 1.7 mm (95% CI 0.72–2.76; *p* = 0.001) was observed in non-ischemic vs. ischemic groups.

### 3.2. Survival Analysis

A time-to-the-event analysis was finally performed, which did not show any evidence of a different risk at follow-up between the two groups, expect for “functional LV improvement (EF increase > 10% baseline)” (HR 3.96, 95% CI: 1.03–15.24; *p* = 0.045) (Figure 2) and TAPSE improvement (HR 0.15, 95% CI 0.04–0.53 *p* = 0.003) (Appendix A) which showed better rates for non-ischemic patients.

We could not find evidence of association with the disease type in the combined endpoint of death and hospitalization for HF (Appendix A Appendix A), possibly due to the low number of events in the follow-up.

We could also observe a greater trend towards a clinically significant improvement in ventricular volumes in non-ischemic patients compared to ischemic patients, but our dataset was probably underpowered to detect the effect (HR: 0.77; 95% CI: 0.36–1.63; *p* = 0.496) (Table 4, Appendix A).

Even the combined-endpoint favorable ventricular remodeling associated with functional improvement (Figure 2, Table 4) showed a favorable trend (log rank test *p* = 0.02) in non-ischemic patients, not reaching the statistical significance after adjusting for age and sex (HR: 0.17; 95% CI: 0.02–1.49; *p* = 0.109). 

A similar outcome was observed in right ventricular systolic-function improvement, measured using TAPSE (Table 4, Appendix A) and considered clinically relevant in cases of more than 10% increase of baseline value. A relevant TAPSE improvement resulted significantly higher in non-ischemic patients (HR 0.15, 95% CI 0.04–0.53 *p* = 0.003).

### 3.3. Factors Predictive of Reverse Remodeling

Finally, through a linear regression model, we analyzed the possible dependence of reverse remodeling, expressed as delta EDV% and calculated as $\frac{\left(\mathrm{EDV}1-\mathrm{EDV}0\right)}{\mathrm{EDV}0}\times 100$, from different factors such as age, sex, time from diagnosis of HF, baseline NT-proBNP, baseline furosemide dose, baseline degree of diastolic dysfunction, baseline LAV, follow-up time and final SV dose, aside with etiology (ischemic/non ischemic). We noticed an increase in the variance explained by the model after the addition of clinically relevant covariates; however, our findings need to be confirmed in a larger dataset (Appendix A).

## 4. Discussion

The current study showed a positive effect of SV in the whole population of HFrEF patients included, with a reduction in EDV and ESV (−15 mL, *p* = 0.0002 and −16 mL, *p* = 0.00005 respectively) and an increase in EF (+3%, *p* = 0.0002) from baseline, which was associated to a reduction in NT-proBNP (−538 pg/mL, *p* = 0.01) and in furosemide dosage (−10 mg/die, *p* = 0.02). An improvement of NYHA functional class was also observed in 56% of the patients (*p* = 0.0002). Recent studies showed an improvement in NYHA class and a decrease in NT-proBNP values after SV treatment and a 5.8% EF improvement in a cohort of 134 patients on SV (EF from 28% ± 5.8% to 31.8% ± 7.3%, *p* < 0.0001) and in a large meta-analysis (average improvement in EF of 5.1% [22]). These results could be influenced by the final SV dose achieved [23].

According to very recent findings from the PARADISE-MI trial, not already published, in patients following acute myocardial infarction, SV did not significantly reduce the rate of CV death, HF hospitalization or outpatient HF requiring treatment, compared with the ACEi ramipril.

To our best knowledge, this is the first prospective study aimed at evaluating a possible different response to SV in a very well selected population of chronic HF and stable disease with dilated cardiomyopathy of ischemic or non-ischemic etiology.

The benefit of SV was more evident in non-ischemic patients, who experienced a significant improvement in EDV, ESV and EF. These results were also confirmed for the combined endpoint “reverse ventricular remodeling associated with improvement in systolic function”, in which the *p* value of the log rank test (0.0215) suggests a favorable trend in non-ischemic patients compared to those ischemic, without however reaching statistical significance, when normalized for age and sex. 

A 10% level of EF improvement was considered clinically significant accordingly to previous studies on reverse remodeling and to our intra-observer variability (3.6%, SD = 3%). Similarly, in consideration of both recent studies on reverse remodeling [24,25] and our intra-observer variability (6.3%, SD = 3.3%), an “EDV improvement” was considered clinically significant if a more-than-10% reduction of baseline levels was noticed. These observations are in line with previous data from the IMPROVE-HF trial that identified non ischemic HF etiology as one of the determinants associated with a >10% improvement in EF [26]. A very recent study conducted in Taiwan [27] showed a marked improvement in EF also in ischemic patients (from 33, SD = 7.5% to 52, SD = 7.4%), but inclusion criteria were different (i.e., EF < 40%) and the study allowed the possibility of revascularizing the patients after their inclusion. 

RV recovery is associated with improved survival in HF patients [28] and TAPSE is an important prognostic marker, either when assessed alone or in combination with pulmonary artery systolic pressure (PASP) [29,30]. In our non-ischemic patients, right ventricular function improved (TAPSE increase: 1.2 mm, *p* = 0.0009), whilst no difference was seen in the ischemic group. Even apparently poor, this improvement is similar to findings obtained in 60 patients with HFrEF of the DAUNIA heart failure registry, where TAPSE changed from 16.5 (4.0) mm to 17.8 (3.9) (*p* < 0.001) [31].

Our data also showed a trend towards a clinically relevant reverse ventricular remodeling (>10% reduction in baseline EDV) in non-ischemic patients compared to ischemic patients, which did not reach the statistical significance (HR 1.3, 95% CI 0.61–2.75, *p* = 0.496. This could be due to the low sample size or the limited follow-up of our study. 

Other studies have found a reverse remodeling in 25–33% of the patients [12,25], which occurred in the first year after SV began and correlated with the absence of myocardial fibrosis in magnetic resonance imaging (CMR) [24,26]. It appears therefore reasonable that patients with HFrEF of non-ischemic etiology, who typically have a lower LGE extension if compared to ischemic patients, would respond better to optimal medical therapy and benefit more from SV therapy. 

Our observations need further confirmation on a larger population analyzed both with echocardiogram and CMR, which could also identify the extension of myocardial fibrosis. 

The challenge of better classifying biological, structural and functional characteristics of ventricular dysfunction, could imply important clinical consequences and influence current indications for defibrillator implantation [5]. 

The combined endpoint of “death from all causes and hospitalization for HF” did not show a different incidence in the two groups of patients, presumably due to the low number of events observed, even if a recent analysis of PARADIGM-HF [15] showed similar results in CV mortality, all-cause mortality and hospitalizations for HF in the two etiological categories (ischemic and non-ischemic heart disease) both on enalapril or SV.

## 5. Limitations

The main shortcoming of the study is the relatively small sample size, the significant gender-related imbalance and the impossibility of a gender-matched analysis. Due to the small number of included patients and differences in the baseline characteristics, it was difficult to obtain more relevant statistical conclusions than those that emerged. 

Other limits could depend on different clinical characteristics of the two groups (i.e., more NYHA III class, diastolic dysfunction, creatinine levels in ischemic patients). 

## 6. Conclusions

The results of our study confirmed that in patients with HF and severe left ventricular dysfunction, SV therapy leads to an overall reduction in end-diastolic and end-systolic volume and to an increase in EF from the baseline. To the best of our knowledge, this is the first study suggesting a greater effect of SV in terms of improvement in right systolic function, left ventricular ejection fraction, and ventricular remodeling in patients with non-ischemic versus ischemic etiology.

These observations need to be confirmed on a larger sample, to better identify the role of SV in the different etiologies of HF, not only in terms of reverse remodeling but also in terms of the important prognostic-therapeutic implications related to it.

## Figures and Tables

**Figure 1 jcm-12-00621-f001:**
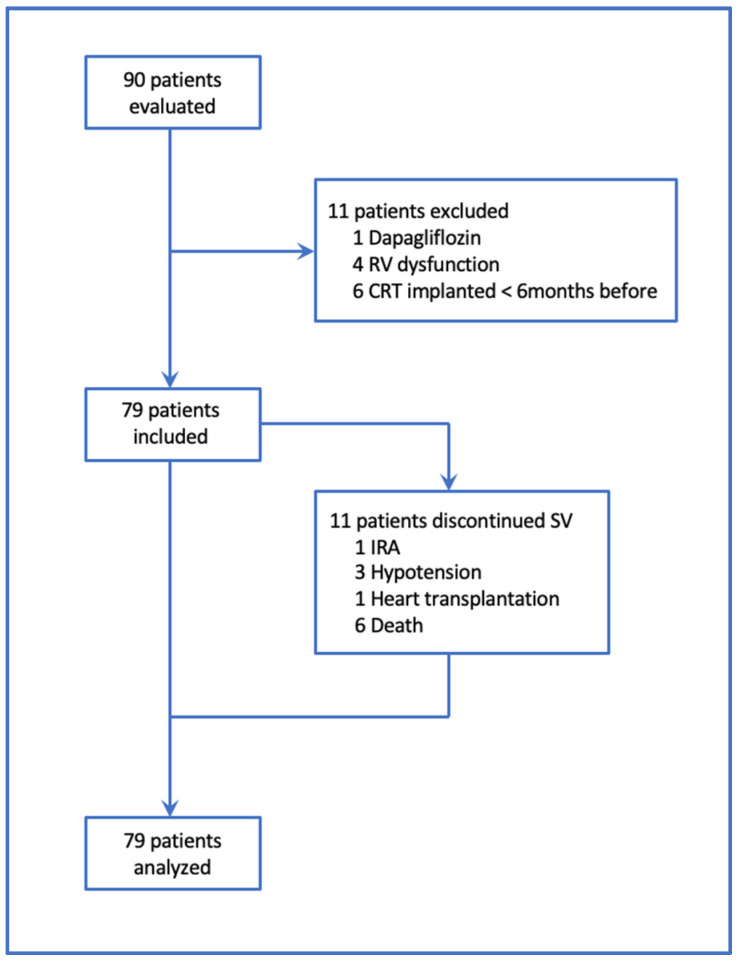
Enrollment of study population. From 90 patients evaluated at the beginning of the study, 11 were rolled out because of exclusion criteria. 79 patients were analyzed, 11 of them discontinued sacubitril/valsartan (SV) before the end of the study.

**Figure 2 jcm-12-00621-f002:**
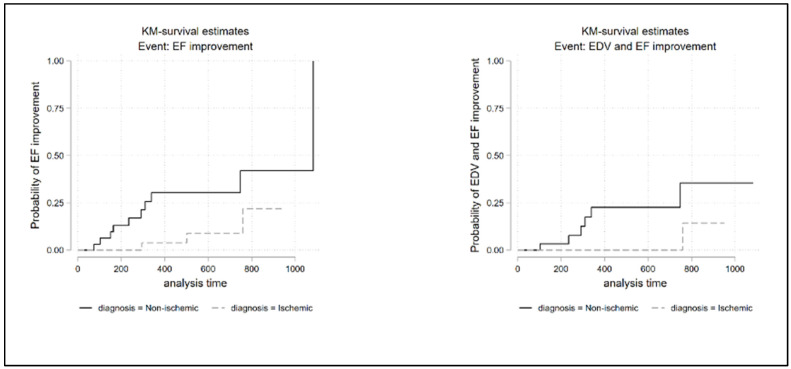
Kaplan–Meier survival analysis and log rank test. Functional LV improvement (EF improvement) and combined outcome of favorable LV remodeling/functional LV improvement (EDV and EF improvement) were significantly higher in non-ischemic group (P log rank test for EF improvement = 0.0224; P log rank test for EDV and EF improvement = 0.0215). LV = left ventricle, EDV = end diastolic volume, EF = ejection fraction.

**Table 1 jcm-12-00621-t001:** Patient’s characteristics at baseline. Categorical variables are represented as frequencies and percentages, continuous variables as median and interquartile range or mean and standard deviation. Ischemic and non-ischemic group are compared using Fisher’s exact test; continuous variables are compared with *t*-test for independent data or Wilcoxon non-parametric test. BMI = body mass index, GFR = glomerular filtration rate, NT-proBNP = N-terminal pro-B-type natriuretic peptide, NYHA = New York Hearth Association, SV = sacubitril/valsartan, HF = heart failure, TIA = transitory ischemic attack, ACEi = angiotensin-converting enzyme inhibitors, ARB = angiotensin receptor blocker, MRA = mineralocorticoid receptor antagonist, ICD = implantable cardioverter defibrillator, CRT = cardiac resynchronization therapy; EDVi = end diastolic volume index, ESVi = end systolic volume index, EF = ejection fraction, LAVi = left atrial volume index, TAPSE = tricuspid annular plane systolic excursion, PAPs = systolic pulmonary artery pressure. Bold values are statistically significant.

	Non-Ischemic	Ischemic	All	*p* Value
	(*n* = 37)	(*n* = 42)	(*n* = 79)	
Age (years)	67 (12)	69 (12)	68 (12)	0.422
Male sex, no. (%)	25 (68)	39 (93)	64 (81)	**0.01**
BMI (Kg/m^2^)	25 (5)	27 (4)	26 (4)	0.222
Heart rate (bpm)	72 (65–82)	70 (59–78)	70 (64–80)	0.183
Systolic pressure (mmHg)	115 (110–130)	110 (100–120)	115 (105–120)	0.193
Diastolic pressure (mmHg)	75 (65–80)	70 (60–75)	70 (60–75)	0.096
Serum creatinine (mg/dL)	1.04 (0.95–1.3)	1.19 (1.06–1.47)	1.11 (0.99–1.4)	**0.011**
GFR (mL/min/1,73 m^2^)	61 (50–83)	59 (44–76)	60 (46–77)	0.178
Kalium (mmol/L)	4.4 (4.2–4.8)	4.4 (4.2–4.7)	4.4 (4.2–4.7)	0.711
NT-proBNP (pg/mL)	4040 (1279–5929)	2445 (1548–5366)	3265 (1548–5493)	0.432
Time from diagnosis > 5 years, no. (%)	23 (62%)	27 (64%)	50 (63%)	0.446
NYHA class, no. (%)				
2	17 (46)	9 (21)	26 (33)	**0.03**
3	20 (54)	33 (79)	53 (67)	
Diastolic dysfunction	12 (32)	11 (26)	23 (29)	**0.019**
1	18 (49)	11 (26)	29 (37)	
2	7 (19)	20 (48)	27 (34)	
3				
Rhythm at enrollment, no. (%)				
Sinus	13 (35)	25 (59)	38 (48)	**0.05**
Atrial fibrillation	10 (27)	4 (10)	14 (18)	
Biventricular Pacing	14 (38)	13 (31)	27 (34)	
Peak SV dosage, no. (%)				
24/26 mg	26 (70)	29 (69)	55 (70)	1
49/51 mg	11 (30)	13 (31)	24 (30)	
Hypertension, no. (%)	19 (51)	28 (67)	47 (60)	0.178
Diabetes, no. (%)	5 (14)	11 (26)	16 (20)	0.262
History of Atrial fibrillation, no. (%)	17 (46)	20 (48)	37 (47)	1
Previous hospitalization for HF, no. (%)	29 (78)	34 (81)	63 (80)	0.787
Previous myocardial infarction, no. (%)	0 (0)	29 (69)	29 (37)	<0.001
TIA/Stroke, no. (%)	3 (8)	2 (5)	5 (6)	0.661
ACEi no. (%)	32 (86)	35 (83)	67 (85)	0.762
ARB, no. (%)	3 (8)	5 (12)	8 (10)	0.717
Digitalis, no. (%)	2 (5)	0 (0)	2 (3)	0.216
Beta blocker, no. (%)	35 (95)	38 (90)	73 (92)	0.679
MRA, no. (%)	21 (57)	26 (62)	47 (59)	0.654
ICD, no. (%)	19 (51)	23 (55)	42 (53)	0.823
CRT, no. (%)	14 (38)	14 (33)	28 (35)	0.814
EDVi (mL/m^2^)	110 (100–136)	124 (104–152)	121 (102–148)	0.12
ESVi (mL/m^2^)	78 (70–96)	88 (73–115)	82 (70–111)	0.168
EF (%)	31 (24–33)	31 (23–34)	31 (23–34)	0.949
LAVi (mL/m^2^)	52 (41–73)	53 (43–68)	53 (41–69)	0.94
TAPSE (mm)	18 (16–20)	18 (16–21)	18 (16–20)	0.711
E/A	1.1 (0.6–2.0)	1.2(0.7–2.6)	1.2(0.6–2.4)	0.206
E/E’	12.4 (10.3–18.7)	14 (10–19)	13.8 (10.3–18.6)	0.743
PAPs (mmHg)	33 (28–44)	37 (29–49)	34 (28–47)	0.385

**Table 2 jcm-12-00621-t002:** Differences in clinical and echocardiographic parameters between follow up and baseline. Means of differences and standard deviations in brackets; the variable “delta NT-proBNP” is characterized by some extreme outliers and therefore it is shown as medians of the differences and interquartile range in brackets. GFR = Glomerular filtration rate, NT-proBNP = N-terminal pro-B-type natriuretic peptide, EDV = end diastolic volume, ESV = end systolic volume, EF = ejection fraction, TAPSE = tricuspid annular plane systolic excursion. Bold values are statistically significant.

Follow-Up—Baseline	Non-Ischemic (*n* = 37)	Ischemic (*n* = 42)	All (*n* = 79)	*p* Value Non-Ischemic (*n* = 37)	*p* Value Ischemic (*n* = 42)	*p* Value All (*n* = 79)
Systolic pressure (mmHg)	−6 (15)	−9 (14)	−7 (15)	**0.0059**	**0.0005**	**9.23 × 10^−6^**
GFR (mL/min/1.73 m^2^)	−1 (12)	−4 (13)	−3 (12)	0.6196	**0.0189**	**0.0439**
NT-pro-BNP (pg/mL)	−571 (−3922; 64)	−251 (−2180; 489)	−538 (−3156; 279)	**0.0236**	0.2129	**0.0159**
Furosemide dosage (mg)	−3 (29)	−15 (46)	−10 (39)	0.4108	**0.018**	**0.0221**
EDV (mL)	−20 (30)	−11 (41)	−15 (37)	**0.0004**	0.0794	**0.0002**
EDVi (mL/m^2^)	−10.87 (16.09)	−5.91 (21.87)	−8.23 (19.42)	**0.0002**	0.0833	**0.0002**
ESV (mL)	−23 (27)	−10 (36)	−16 (33)	**4.00 × 10^−6^**	0.0515	**5.64 × 10^−6^**
ESVi (mL/m^2^)	−12.58 (14.59)	−5.18 (19.35)	−8.65 (17.57)	**3.19 × 10^−6^**	0.0527	**4.20 × 10^−6^**
EF (%)	6 (8)	1 (6)	3 (7)	**3.26 × 10^−5^**	0.2555	**0.0002**
TAPSE (mm)	1.19 (1.96)	−0.33 (2.3)	0 (2)	**0.0009**	0.4863	0.0736
E/E’	1.39 (5.2)	−0.13 (8.2)	0.55 (7.02)	**0.0497**	0.5156	0.0763

**Table 3 jcm-12-00621-t003:** Results of linear regression. A significant improvement of EF and TAPSE was observed in non-ischemic vs ischemic groups. The beta coefficient expresses the increase (if >0) or decrease (if <0) estimated for each delta, moving from the non-ischemic group (reference category for comparison) to ischemic. SE = standard error, CI = confidence interval, SBP = systolic blood pressure, GFR = Glomerular filtration rate, NYHA = New York Hearth Association, EDV = end diastolic volume, ESV = end systolic volume, EF = ejection fraction, TAPSE = tricuspid annular plane systolic excursion. Bold values are statistically significant.

Follow-Up—Baseline	Beta (SE)	95% CI	*p*
Endpoints			
SBP (mmHg)	−0.73 (3.2)	−7.2; 5.7	0.842
GFR (mL/min)	−3.22 (2.9)	−9.1; 2.6	0.276
NT-proBNP (pg/mL)	1134 (1674)	−2227; 4496	0.501
Furosemide dosage (mg)	−11 (8.7)	−28.5; 6.3	0.208
NYHA	−0.2 (0.14)	−0.5; 0.05	0.108
Diastolic dysfunction	0.005 (0.14)	−0.28; 0.27	0.974
E/E’	−1.27 (1.55)	−4.4; 1.8	0.415
Other parameters			
EDV (mL)	8 (0.7)	−9; 25.8	0.34
ESV (mL)	14 (7.5)	−1.08; 29	0.068
EF (%)	−6 (1.7)	−9; −2.4	**0.001**
TAPSE (mm)	−1.7 (1.6)	−2.75; −0.72	**0.001**

**Table 4 jcm-12-00621-t004:** Results of Cox regression. Functional LV improvement and TAPSE improvement were significant in non-ischemic group. Combined endpoint of EDV and EF improvement was non-significant but showed a positive trend for non-ischemic group. Reference category for HR computation was the non-ischemic group, apart from the functional LV improvement endpoint, where reference category was the ischemic group. EDV = end diastolic volume, ESV = end systolic volume, EF = ejection fraction, HF = heart failure; LV = left ventricular, NT-proBNP = N-terminal pro-B-type natriuretic peptide, NYHA = New York Hearth Association, TAPSE = tricuspid annular plane systolic excursion. Bold values are statistically significant.

Endpoints	HR (95% CI)	*p*
Favorable reverse remodeling (EDV reduction >10% baseline)	0.77 (0.36–1.63)	0.496
Functional LV improvement (EF increase > 10% baseline)	3.96 (1.03–15.24)	**0.045**
Combined endpoint of EDV and EF improvement	0.17 (0.02–1.49)	0.109
TAPSE improvement	0.15 (0.04–0.53)	**0.003**
Decrease in NT-proBNP > 326 pg/mL	0.76 (0.34–1.65)	0.483
NYHA improvement	1.45 (0.73–2.88)	0.289
Death or hospitalization for HF	0.63 (0.23–1.73)	0.368
Furosemide dosage reduction >10% baseline	1.04 (0.44–2.48)	0.921

## Data Availability

Data supporting results could be asked at the authors but only anonymously.

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
