# Peer review of "Reverse Remodeling and Functional Improvement of Left Ventricle in Patients with Chronic Heart Failure Treated with Sacubitril/Valsartan: Comparison between Non-Ischemic and Ischemic Etiology"

_jcm, 2023, doi:10.3390/jcm12020621_

Round 1

Reviewer 1 Report

unlike what the authors stated, these works are found in the literature which have the purpose of differentiating the effect of sacubitrilvalsartan based on the ischemic or non-ischemic etiology of the heart failure: Ioannou, A., Metaxa, S., Simon, S., Mandal, A. K. J., & Missouris, C. G. (2020). Comparison of the Effect of Sacubitril/Valsartan on Left Ventricular Systolic Function in Patients with Non-ischaemic and Ischaemic Cardiomyopathy. Cardiovascular drugs and therapy34(6), 755–762.

Lee, Y. H., Chiou, W. R., Hsu, C. Y., Lin, P. L., Liang, H. W., Chung, F. P., Liao, C. T., Lin, W. Y., & Chang, H. Y. (2022). Different left ventricular remodelling patterns and clinical outcomes between non-ischaemic and ischaemic aetiologies in heart failure patients receiving sacubitril/valsartan treatment. European heart journal. Cardiovascular pharmacotherapy8(2), 118–129.

Abumayyaleh, M., Pilsinger, C., El-Battrawy, I., Kummer, M., Kuschyk, J., Borggrefe, M., Mügge, A., Aweimer, A., & Akin, I. (2021). Clinical Outcomes in Patients with Ischemic versus Non-Ischemic Cardiomyopathy after Angiotensin-Neprilysin Inhibition Therapy. Journal of clinical medicine10(21), 4989.

The article should be rewritten in light of the previous literature

Author Response

Referee 1#

  • We added the suggested bibliography and rewrote the introduction according to it (lines 52-55)

Reviewer 2 Report

Article
Reverse remodeling in patients with non-ischemic chronic heart failure treated with sacubitril/valsartan

Heart failure syndrome was first described as an emerging epidemic more than two decades ago. Today, due to a growing and aging population, the total number of patients with heart failure continues to rise. The authors deal with a timely and clinically relevant topic.

Article interesting, but needs improvement:

1.      Each legend should clearly describe what is being shown in the Figure and Table, so that the reader can understand the Figures and Table when looking at them individually and separate from the main manuscript.

2.      Please describe in detail by what criteria left ventricular diastolic function was evaluated.

3.      Please provide the approval number of the local bioethics committee.

4.      Please provide detailed inclusion criteria regarding left ventricular ejection fraction and symptoms.

A limitation of the work is the small size of the groups, which the authors emphasized in the limitations of the work. In addition, a more reliable tool for evaluating the left ventricle is cardiac MRI.

The concept of the work is interesting, despite the limitations it is worth publishing after revisions.

Author Response

Referee 2#

  • Added figure legends (lines 75-77, 163-171, 207-211, 231-235, 246-249, 263-268)
  • Added criteria of left ventricular diastolic dysfunction (lines 83-88)
  • A formal approval of the protocol from the local Ethics Committee was considered unnecessary because study did not interfere with the usual clinical routine, the assessment of cardiac function and laboratory tests were part of the routine examinations prescribed to patients, no change in routine dose administration of the drugs was implemented based on the study results and data were handed anonymously. We have already explained it at thebeginning but we could add the explanation also in the text if needed
  • Added a detailed description of inclusion criteria (lines 66-73)

Reviewer 3 Report

The manuscript entitled " Reverse remodeling in patients with non-ischemic chronic heart failure treated with sacubitril/valsartan" by Roberto Cemin et al. evaluates the long-term different effects of sacubitril/valsartan among patients with chronic heart failure.

The strengths of this article: it was added some interesting information in the population of patients with Heart failure and especially in those treated with sacubitril/valsartan.

The weaknesses of the article: the two groups were compared through the etiology of Heart Failure, taking into account the ischemic or non-ischemic reasons.

However, the two groups of patients were not matched by gender.

Since the main comparison is focused on the different effects of S/V, it seems important to match the two groups not only by age, but also by gender.

The title of the article doesn’t reflect fully the contain of the manuscript.

There are some minor problems in the text of abstract such as:
In page 1, in the text of Abstract:  "69 patients»; please correct a number of included patients.

The aim of the article in the abstract is different from the aim in the introduction.

Methods. The authors wrote that 11 patients discontinued SV (in a Figure 1). Please, clarify what does it mean (at the begining of the study, over which time interval? What about adjudication of hospitalizations/deaths?)

Moreover the authors should give the precise recommendations.

The list of references shall be updated – only 48% of articles were published less than 5 years ago

Author Response

Referee 3#

  • The lack of a match by gender is a design limitation and we cannot fix it at the current stage (we consulted our statisticians). To overcome this problem we adjusted the survival analyses for sex and age (that is still advisable even in presence of an age-matched design). Moreover, we added a statement about the lack of a match by gender in the “Limitations” section (lines 347-350).
  • Title edited in: “Reverse remodeling and functional improvement of left ventricle in patients with chronic heart failure treated with sacubitril/valsartan: comparison between non-ischemic and is-chemic etiology”
  • Corrected the number of patients in the text of abstract
  • Corrected the aim of the article in both abstract and full text
  • Clarifications about the 11 patients who discontinued SV (lines 131-145)
  • Updated bibliography as suggested (65% of citations published within the last 5 years)

Round 2

Reviewer 1 Report

The paper has improved after review by the authors. I appreciate this version.

Reviewer 2 Report

I thank the authors for taking my suggestions into account when improving the manuscript.
I accept the corrections made.
I have a minor comment:
- "LA vol. > 34 ml/m2" should be "LA vol. index> 34 ml/m2".

Reviewer 3 Report

Thank you for the updated information. The manuscript has been sufficiently improved and can be published in JCM.